# A Social Network Analysis of Tweets Related to Masks during the COVID-19 Pandemic

**DOI:** 10.3390/ijerph17218235

**Published:** 2020-11-07

**Authors:** Wasim Ahmed, Josep Vidal-Alaball, Francesc Lopez Segui, Pedro A. Moreno-Sánchez

**Affiliations:** 1Newcastle University Business School, Newcastle University, Newcastle upon Tyne NE1 4SE, UK; 2Health Promotion in Rural Areas Research Group, Gerència Territorial de la Catalunya Central, Institut Català de la Salut, 08272 Sant Fruitós de Bages, Spain; jvidal.cc.ics@gencat.cat; 3Unitat de Suport a la Recerca de la Catalunya Central, Fundació Institut Universitari per a la Recerca a l’Atenció Primària de Salut Jordi Gol i Gurina, 08272 Sant Fruitós de Bages, Spain; 4TIC Salut Social, Generalitat de Catalunya, 08005 Barcelona, Spain; francesc.lopez@cmail.cat; 5Center for Research in Health and Economics (CRES-UPF), Universitat Pompeu Fabra, 08002 Barcelona, Spain; 6School of Health Care and Social Work, Seinäjoki University of Applied Sciences, 60100 Seinäjoki, Finland; pedro.morenosanchez@seamk.fi

**Keywords:** COVID-19, coronavirus, twitter, masks, transmission, public health

## Abstract

Background: High compliance in wearing a mask is a crucial factor for stopping the transmission of COVID-19. Since the beginning of the pandemic, social media has been a key communication channel for citizens. This study focused on analyzing content from Twitter related to masks during the COVID-19 pandemic. Methods: Twitter data were collected using the keyword “mask” from 27 June 2020 to 4 July 2020. The total number of tweets gathered were *n* = 452,430. A systematic random sample of 1% (*n* = 4525) of tweets was analyzed using social network analysis. NodeXL (Social Media Research Foundation, California, CA, USA) was used to identify users ranked influential by betweenness centrality and was used to identify key hashtags and content. Results: The overall shape of the network resembled a community network because there was a range of users conversing amongst each other in different clusters. It was found that a range of accounts were influential and/or mentioned within the network. These ranged from ordinary citizens, politicians, and popular culture figures. The most common theme and popular hashtags to emerge from the data encouraged the public to wear masks. Conclusion: Towards the end of June 2020, Twitter was utilized by the public to encourage others to wear masks and discussions around masks included a wide range of users.

## 1. Introduction

Since emerging in China in December 2019, the beta coronavirus SARS-CoV-2 (named COVID-19) has been expanding rapidly throughout the world [1]. On 30 January 2020, the World Health Organisation (WHO) declared coronavirus COVID-19 to be a Public Health Emergency of International Concern [2]. Thereafter, in March 2020, the WHO declared COVID-19 a pandemic due to the identification of more than 118,000 cases in 114 countries [3]. Millions of people worldwide have been infected, and hundreds of thousands of people have lost their lives due to COVID-19 [4].

Different regional approaches related to the use of face masks to mitigate the transmission of COVID-19 have been developed. In East Asian countries, for example, wearing masks was ubiquitous and was performed as a hygienic habit due to past positive outcomes in 2003 during SARS. On the contrary, in Europe and North America, the population was informed that masks were not recommended for general use [5]. The WHO states that masks can be used either for protection of healthy persons (worn to protect oneself when in contact with an infected individual) or for infection control (worn by an infected individual to prevent onward transmission) [6]. Ma et al. showed that N95 masks, medical masks, and even home-made masks could block at least 90% of the virus in aerosols [7]. From the perspective of disease spread, at the population level, wearing masks by infected individuals may be important in helping retain contagious droplets, aerosols, and particles that can infect others and contaminate surfaces [5].

The WHO notes that health workers and caregivers in clinical areas must continuously wear medical masks where there is known or suspected community transmission [3]. However, due to the lack of robust evidence in clinical trials, the WHO’s recommendations about wearing masks by the general population has been ambiguous. In its interim guidance issued on 5 June 2020, the WHO advises that to effectively prevent COVID-19 transmission in areas of community transmission, governments should encourage the public to wear masks.

Universal masking, as a public health intervention, would probably intercept the transmission of COVID-19. This could especially be the case for asymptomatic infected individuals with high viral load at the early stage of the disease (suggested to be around 40–80% by Javid et al. [5]) [8]. Therefore, community-wide mask usage irrespective of symptoms may reduce the infectivity of silent asymptomatic individuals. Masks are helpful for source control of asymptomatic infectious persons but also for protecting healthy people [9]. Universal masking may become the default solution in high-risk areas with a large number of patients and without sufficient testing, where everyone can only be seen as potentially infected [10].

High compliance in wearing a mask is a crucial factor for stopping transmission. This is similar to vaccines: the more people that are vaccinated, the higher the benefit to the whole population, including those who cannot be vaccinated, such as infants or immune-compromised people [11]. Moreover, the gain is greater the earlier masks are adopted and when face masks are used to complement other measures such as social distancing [12]. Hand hygiene is a discontinuous process and sometimes difficult to practice in the community. However, wearing a mask is a continuous form of protection to stop respiratory droplets to and from others [13]. Thus, controlling harms at the source by wearing masks is at least as important as other mitigation actions such as handwashing. Universal masking would also help in removing stigmatization that could discourage symptomatic patients to wear a mask in many places, preventing any discrimination that might arise.

However, various authors have justified not wearing masks for different reasons. One of the key reasons is that there is limited evidence supported by clinical trials on the effectiveness of masks [14]. Secondly, it is claimed that prevention depends on an individual’s behaviour and compliance, which has been shown to be inconsistent or inappropriate in trials, for example, people may repeatedly touch their mask [6]. It is also claimed that wearing a mask might make people feel safe and hence reduce adherence to other nonpharmaceutical measures such as hand washing and social distancing [5]. Moreover, at one stage it was also argued that because of the shortage of masks, the public should not wear them because healthcare workers would need them more and that the public buying masks could lead to major supply chain problems along with an increase in prices [15].

In this context, the aim of this study was to look at potential for Twitter to highlight public views towards universal masking. Social media is a useful platform for raising awareness of various issues and Twitter is a valuable platform for listening to public views and opinions on a range of topics in real-time. Moreover, from a public health standpoint, it is important to develop an understanding of the drivers of the discussion around masks and to gain insight into key topics of discussion. 

More specifically, the research questions of this study were as follows: What was the overall network shape of the discussion on Twitter?What were the key hashtags?Who were the most influential users?Who were the most mentioned users?What were the key themes of discussion that were taking place?

## 2. Methods 

### 2.1. Tweet Sampling

Twitter data were collected using the keyword “mask” from 27 June 2020 21:21:21 to 4 July 2020 19:50:40 to provide coverage of about a week of data in a period when this topic was highly present in social media. The total number of tweets which were gathered (worldwide) were *n* = 452,430. By using the keyword “mask”, tweets including words such as “masks” or “#mask” were also retrieved and included. A systematic random sample of 1% (*n* = 4525) of the tweets was extracted and analyzed using social network analysis, as described in the next section.

### 2.2. Social Network Analysis

The software NodeXL (Social Media Research Foundation, California, CA, USA) was used to conduct a social network analysis of the data [16]. In understanding the network graph, the results of this study build upon previous research [17,18], which has highlighted that Twitter topics may follow six network shapes and structures [19]: broadcast networks, polarized crowds, brand clusters, tight crowds, community clusters, and support networks. In the network graph provided in Figure 1, circles represent individual Twitter users and the lines between them represent connections such as mention and reply. The network graph was laid out using the Harel–Koren Fast Multiscale layout algorithm which is built into NodeXL. NodeXL uses the Search Application Processing Interface (API). Influential users were identified using NodeXL and were anonymized by providing a description of the account in line with previous research [18]. A specific subanalysis was performed for tweets that originated solely from the USA. Regional information was extracted as follows: a total of *n* = 13,265 tweets were extracted where users had included “USA” in their user bios and a 5% sample of tweets was extracted and analyzed in NodeXL. Individual users and/or organisations that were deemed to be not sufficiently in the “public domain” were anonymized. 

## 3. Results

### 3.1. Top 10 Hashtags Used

Table 1 below provides an overview of the most used hashtags during this time, showing that “wearamask” (*n* = 34) and “maskssavelives” (*n* = 11) appeared among the most used hashtags. 

The hashtag “kidlitformasks” referred to authors of children’s books who used this hashtag to highlight the importance of wearing masks. Twitter users also shared images from a Japanese manga series “haikyuu” and at the same time encouraged others to wear a mask using the “#haikyuu” and ‘hq’ hashtags, and these appeared as the 7th and 8th most popular hashtags that were used. As the discussion during this time was global in nature, there also appeared to be relevant hashtags from around the world such as “マスク” which refers to “mask” in Japanese. The word “ビール” also appeared, which is the Japanese word for “beer”. This was because users conversed about a humorous mask invention which allows a user to store beer inside the mask which can be consumed. The hashtag “andhrapradesh” appeared as a popular hashtag as this referred to a state in the south-eastern coastal region of India and news reports were shared at this time related to reports about the death of a young person who was allegedly “beaten to death” by police for not wearing a mask. 

### 3.2. Overview of Network Structure 

Figure 1 is a social network graph of the discussion taking place during this time. The largest group (group 1) is an isolates group which contained users who sent tweets that did not contain mentions. Overall, the group resembled a community network shape because there were many groups of users conversing about this topic. NodeXL clustered users into different groups based on mentions. A community cluster indicated that many users were talking to each other across several groups. It was also interesting to see influential users (indicated by larger circles) scattered around the network, which indicated that the topic brought in a wide range of influential actors. 

### 3.3. Top 10 Users Ranked by Betweeness Centrality 

Table 2 highlights Twitter users that were influential during this time. The users in the network were ranked by betweenness centrality, which is a measure of centrality. These users would have acted as important bridges within the network. The follower’s column refers to the amount of followers each user has. Many of the influential nodes within the network derived from ordinary citizens who became important bridges in the network. A number of popular culture figures also appeared among the most influential users within the network.

### 3.4. Top 10 Users Most Mentioned

Table 3 provides an overview of users that were most mentioned during this time which ranged from political figures, organizational accounts, and popular culture related accounts. Certain users may not have tweeted about masks but were mentioned frequently by other users. Examining the influential users from this time highlighted that the discussion had been focused in and around the United States. However, it must be noted that not all account mentions may have been relevant to medical masks, as the band Slipknot are known to wear non-medical masks. 

### 3.5. Content Analysis 

NodeXL was used to identify the most frequently occurring co-words, as shown in Table 4. These co-related keywords provided insight into the types of discussions that may have been taking place on Twitter. Word pairs containing mentions of Twitter user handles were removed. This occurred on three occasions in group 3. 

The most frequent co-related keywords were “wear” and “mask” to form the sentence “wear a mask” and, in some cases, also involved an expletive. There were also tweets related to humor. There were also co-occurring words such as “stay” and “home”. Other co-occurring words which appeared across the clusters included “breathing” and “problem”, which revolved around the debate of whether face masks should be mandatory as they may lead to breathing problems. In regard to this debate, Twitter users provided evidence to highlight that face masks did not cause breathing problems, whereas other users highlighted the potential for masks to cause breathing issues. 

### 3.6. Regional Analysis of the USA

In this part of the analysis, tweets were extracted to only focus on the USA. Data were filtered by users who noted in their bios that they were from the USA.

#### Top Word Pairs

Table 5 provides an overview of the 5 most popular words that were used from users from the USA.

The most frequent words used together included “wear mask”, “wearing mask”, “mask, public”, “face, mask”, and “breathing problem”. The first two co-words appeared to be encouraging the use of face masks and the third and fourth most used co-words appeared to be centred around general discussions around the use of masks. The fifth most used co-word appeared to relate to discussions around whether masks could cause breathing problems. Overall, there appeared to be overlaps between phrases and words used in tweets when filtering specifically for the USA, compared to analyzing tweets overall. Furthermore, content such as top hashtags and users appeared to be similar to that of the results of the analysis overall.

## 4. Discussion

The overall shape of the network resembled a community as there were a range of users conversing amongst each other in different clusters. When examining the most frequently used hashtags, it appeared that the most popular hashtags encouraged mask wearing among the public. It was found that a range of accounts were influential and/or mentioned, ranging from ordinary citizens, politicians, and popular culture figures. The discussion had been politicized by some users on Twitter, which led to politicians appearing as influential users within the network. Japanese hashtags also appeared within the network, which highlights the global nature of the discussions around this issue. It is important to note that some of the accounts such as the official Twitter account of YouTube might not have tweeted using the word ‘mask’, however, users may have shared content which contained the word ‘mask’ and also used the mention “@YouTube”. The most common theme to emerge was the encouragement for the public to wear masks and discussions around this. Other themes were related to jokes and discussions related to whether face masks were safe for those who may have breathing problems. Our study also examined Twitter data emerging solely from the United States, which demonstrated that there was overlap in content. This could have occurred because the Twitter has the most active users in the United States.

It must be noted that Twitter discussions are constantly evolving and potentially alter on a weekly basis. A limitation of our study is that it focused specifically on the Twitter network from 27 June to the 4 July, hence our findings may not be applicable to other time periods. Future research could seek to expand time periods and examine Twitter discussions based on other locations. A further limitation is that with the 1% random sample we extracted, generic keyword and 7-day approach, we have captured some tweets coming from “temporary” discussions (for instance, beer and mask related humor in Japan). If the study had examined a longer time period, i.e., 6 months, and taken a 0.005% sample, the study would have been able to perform a much wider analysis (the masks and beer humor would not have been captured and/or captured on a lesser scale). At the same time, however, the approach adopted in this study had a better chance of capturing other issues such as conspiracy theories and/or short-lived time-based discussions. Future research could combine both approaches in the analysis. Further research could also seek to conduct a sentiment analysis of the data. A wider limitation related to research on Twitter is the potential of ‘off-topic’ discussions surrounding a particular keyword or hashtag to take place. Future research could seek to eliminate irrelevant content prior to analyzing data. Twitter data can be used to study a wide range of public health topics and was recently used to study views into personal health records [20], which utilises a similar methodology to this present study. Other research has also examined disclosure of patient information [21] and COVID-19 conspiracies [22]. Future research could also seek to examine the role of influential accounts during this time.

## 5. Conclusions

Overall, it was found that the shape of the network resembled a community as there were a range of users conversing amongst each other in different clusters. It was found that a range of accounts were influential and/or mentioned ranging from ordinary citizens, politicians, and popular culture figures. The most common theme and popular hashtags to emerge from the data encouraged the public to wear masks. Public health authorities and influential accounts could continue to utilize social media platforms to encourage users to wear masks. 

## Figures and Tables

**Figure 1 ijerph-17-08235-f001:**
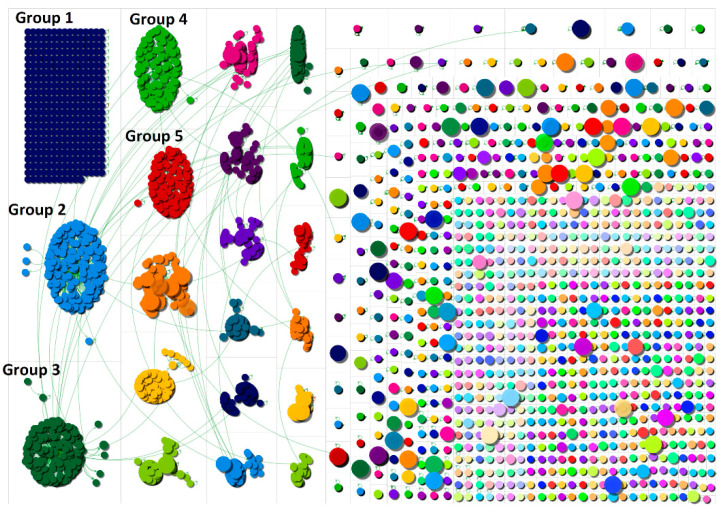
Network graph of masks from 27 June to 4 July.

**Table 1 ijerph-17-08235-t001:** Top 10 hashtags.

Rank	Top Hashtags in Tweet in Entire Graph	Entire Graph Count
1	covid19	50
2	wearamask	34
3	kidlitformasks	23
4	mask	18
5	maskssavelives	15
6	マスク (Japanese for “mask”)	13
7	haikyuu	11
8	hq	11
9	ビール (Japanese for “beer”)	10
10	andhrapradesh	9

**Table 2 ijerph-17-08235-t002:** Top 10 users ranked by betweenness centrality.

Rank	Account Type	Followers
1	Citizen	301
2	Citizen	2434
3	Citizen	3160
4	Citizen	206
5	Professor	4602
6	Citizen	2601
7	Citizen	17
8	Citizen	1002
9	Rex Chapman also known as Ice-T (American rapper)	1,786,850
10	Rex Chapman (former basketball player)	1,004,702

**Table 3 ijerph-17-08235-t003:** Top 10 users most mentioned.

Rank	Account Type	Followers
1	Donald Trump (current president of the United States)	83.8 Million
2	Joe Biden (former Vice President of the United States)	7.1 Million
3	Account Belonging to an Organization	23.1 Thousand
4	Bangtan Boys, a seven-member South Korean boy band	27.2 Million
5	Mike Pence (Vice President of the United States)	9.3 Million
6	Sarah Silverman (USA based comedian)	12.4 Million
7	Official Twitter account of YouTube	72.1 Million
8	Account Belonging to an Organisation	2.2 Thousand
9	Citizen	20.1 Thousand
10	Slipknot (an American heavy metal band from Des Moines)	2.1 Million

**Table 4 ijerph-17-08235-t004:** Identifying content across clusters.

NodeXL Group	Frequent Word Pairs
1	‘Wear, mask’ ‘wearing, mask’ ‘face, mask’ ‘f***ing, mask’ ‘wear, f***ing*’ ‘mask, public’ ‘stay, home’
2	‘s**t*, mask,’ ‘oh, s**t*’ ‘mask, really’ ‘really, thing’ ‘Wearing, mask’ ‘convince, y’all’ ‘y’all, wearing’ ‘Mask, hurt’ ‘hurt, dog’ ‘dog, finally’
3	‘Wear, mask’ ‘breathing, problem’ ‘problem, lol’ ‘Lol, wear’ ‘mask, karen’ ‘f***ing*, hard’ ‘Hard, wear’
4	‘Someone, ugly’ ‘ugly, personally’ ‘personally, mind’ ‘Mind, wearing’ ‘wearing, mask’ ‘mask, covers’ ‘Covers, half’ ‘half, face’ ‘wear, mask’ ‘breathing, problem’
5	‘Wear, mask’ ‘see, someone’ ‘wearing, mask’

* asterisk placed by authors in expletives.

**Table 5 ijerph-17-08235-t005:** Top 5 word pairs. Filtered by USA.

Rank	Word Pair
1	‘wear, mask’
2	‘wearing, mask’
3	‘mask, public’
4	‘face, mask’
5	‘breathing, problem’

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
