# Peer review of "A Social Network Analysis of Tweets Related to Masks during the COVID-19 Pandemic"

_ijerph, 2020, doi:10.3390/ijerph17218235_

Round 1

Reviewer 1 Report

A social network analysis of tweets related to masks and COVID-19

The topic is current and could be interesting if it had solid theoretical and methodological foundations. The reliability of the results, in relation to the reproducibility and accuracy of the research (Krippendorff, 2004), is doubtful. I cannot recommend the publication of this version of the article, because I cannot see what the contribution of this research is to the theoretical and methodological field of public health.

Readers may raise some questions:

Why did you select that particular week?

Why did they select USA to do a subanalysis?

Why did you choose NodeXL? How reliable is this software?

The article raises many doubts:

In relation to the figures you give, I do not understand that the sample of 45,243 tweets is 1% of 452,430.

The explanation in the graph in Figure 1 is wholly inadequate.

In the explanation in Table 2, I do not understand how citizen 7th, with 17 followers, can be an important bridge within the network, compared to other users who have thousands or millions of followers.

"A further limitation is that with the 1% random sample and 1-month approach ...": From what I have read before, I think the current proposal is a 10% sample and that the period is one week.

In the conclusions, there are a couple of comments. The first is shallow. The last sentence is not well based on the results of the investigation. The theoretical and methodological implications, as well as the implications for public health managers, are lacking.

Author Response

Reviewer 1

Reviewer Comment

Author Reply

Readers may raise some questions:

    •  Why did you select that particular week?
    • Why did they select USA to do a subanalysis?
    • Why did you choose NodeXL? How reliable is this software?

This has been explained in more detail in the methods section.

The software NodeXL has been used in several other published studies and it is a reliable and recognised tool to analyse twitter data.

 The article raises many doubts:

 In relation to the figures you give, I do not understand that the sample of 45,243 tweets is 1% of 452,430.

This is a typo as the figure is 4524.3 and was rounded up to 4525, this has now been addressed.

The explanation in the graph in Figure 1 is wholly inadequate.

 In the explanation in Table 2, I do not understand how citizen 7th, with 17 followers, can be an important bridge within the network, compared to other users who have thousands or millions of followers.

This has been explained in more detail.

 "A further limitation is that with the 1% random sample and 1-month approach ...": From what I have read before, I think the current proposal is a 10% sample and that the period is one week.

This was a typo and has now been resolved.

 In the conclusions, there are a couple of comments. The first is shallow. The last sentence is not well based on the results of the investigation. The theoretical and methodological implications, as well as the implications for public health managers, are lacking.

We have added some specific recommendations in the conclusion.

Reviewer 2 Report

The authors of this paper were interested in investigating the use of Twitter in relation to the spreading of information regarding face masks. In order to focus their research into an understandable and directed manner they proposed 5 key research questions. These were:

  1. What was the overall network shape of the discussion on Twitter?
  2. What were they key hashtags?
  3. Who were the most influential users?
  4. Who were the most mentioned users?
  5. What were the key themes of discussion that were taking place?

By investigating these questions, the authors hoped to be able to determine the nature of the discussion on Twitter in regards to face masks. The authors collected Tweets over a week at the end of June 2020. They collected a little over 450,000 Tweets. From these collected Tweets the authors randomly sampled 1%. In addition, the authors were interested in examining the United States in particular as well. To do this they selected all Tweets whose user had USA in their bio. This resulted in 13,265 Tweets. They sampled 5% of these to determine their results.

Overall, I think authors did a good job. I do have some comments as follows:

  1. The explanation of the significance of the fact that the discussion formed a “community” shape need to be explained more. Authors discovered this and then offer no insight into what this means really.
  2. Going into more detail on the roles that the influential people actually played could have been interesting. Authors have this list of 10 people but we do not know what they really did. Where they connecting a number of different groups together? How much were each of them posting? What was the general content of each post? Questions like these would be interesting to have answers to.
  3. The authors conclusions seem somewhat hand wavy. They have several general conclusions that are not overly committed a lot.

Author Response

Reviewer 2

Reviewer Comment

Author Reply

The explanation of the significance of the fact that the discussion formed a “community” shape need to be explained more. Authors discovered this and then offer no insight into what this means really.

This has been explained in more detail.

Going into more detail on the roles that the influential people actually played could have been interesting. Authors have this list of 10 people but we do not know what they really did. Where they connecting a number of different groups together? How much were each of them posting? What was the general content of each post? Questions like these would be interesting to have answers to.

This has been noted in the future research section.

The authors conclusions seem somewhat hand wavy. They have several general conclusions that are not overly committed a lot.

We have added some specific recommendations in the conclusion.

Reviewer 3 Report

Line 52: Change “somehow” to “somewhat”

Line 57: This entire paragraph needs some rewording. Some words are grammatically in the wrong place (“Not only masks are helpful” should be “Not only are masks helpful”. No comma after universal masking in the first sentence. The point of the paragraph is clear but it reads clunky.

Line 65: This paragraph has extra words and is also kindof clunky. I understand the point, but there is a lot of repetition. I think it could be cut way down and combined with the next paragraph.

Line 77: Both this paragraph and the one above seem kindof out of place. I understand emphasizing the importance of masking and policies surrounding it, but you spend too much time establishing the importance and not enough time on the next paragraph, tying together Twitter use and masking. It feels disjointed.

Line 86: The final paragraph should be longer. I like using numbered points, but they should be incorporated into the text better, rather than a separate list. You could even put them as a table in the methods

Line 96: The entire methods section should probably be separated into at least 2 subsections for clarity. I.e. Tweet Sampling, Social network analysis. It’s overall a little wordy for a methods paragraph.

Line 106: “In the network circles” probably needs hyphens for clarity

Line 116: Results 3.1: I don’t think you need the first section at all, the top 10 hashtags. It seems pointless, especially explaining the subtle cultural contexts for some of the tags, like beer. I would drop this section entirely.

Line 143: This section needs some major work. First off, structure is spelled wrong in the title. The first sentence is not needed. The main problem is the text is just talking about the figure, as opposed to the figure supporting the text. It also needs way more analysis of the groups in the figure. The entire paragraph is really generic and feels pointless without more analysis. The actual analysis is cool with a cool figure, get into it!

Line 151: There is a decided lack of statistical methods for all of these figures. You didn’t define “bridges” in the context of followers. Also, how are you measuring influence? Just because you have followers doesn’t mean you actually influence them. Maybe measuring retweets would be better, and maybe you did do that, but it’s not clear if you did. You also didn’t define betweenness centrality. What is that?

Line 159: This paragraph feels very unscientific and pointless. Too many “suggests”, “may” and “might”. It just doesn’t feel informative about, well, anything, it’s just a list of twitter followers.

Line 169: Once again, not much here to help the reader interpret the data.

Line 179: “revolved” not “resolved”

Line 181: This last sentence is just unsupported by any data presented. I get that it’s providing some clarity, but its author interpretation, so it should be in the discussion.

Line 187: How were these ranks determined? “The content…appeared to be similar to that of the results of the analysis overall”, How so? What criteria are you using to determine this?

Line 206: Lose the first sentence, and you don’t need to introduce each topic being discussed, i.e. “in response to the first research question”. The discussion should go back over the main questions and hypotheses from the intro, then go back through the results to show how they addressed those questions, and provide some interpretation. Your limitations are noted, and you mention only looking at such a limited weekly sample. I agree, that is a big limitation. This pandemic, and the politics behind it, was incredibly dynamic and changing every day, let alone over a week or month.

Line 236: This paragraph is incomplete and has multiple grammatical and spelling errors. Also, your last reference needs to be updated in the bibliography, it was published August 18.

Overall this needs some work, especially with the results. I would dig into the nodes more, and talk more about the influences of these “bridges”. Maybe even pick a specific twitter account or two and try to follow information as it moves.

Author Response

Reviewer 3

Reviewer Comment

Author Reply

Line 52: Change “somehow” to “somewhat”

This has been done.

Line 57: This entire paragraph needs some rewording. Some words are grammatically in the wrong place (“Not only masks are helpful” should be “Not only are masks helpful”. No comma after universal masking in the first sentence. The point of the paragraph is clear but it reads clunky.

This paragraph has been altered.

Line 65: This paragraph has extra words and is also kindof clunky. I understand the point, but there is a lot of repetition. I think it could be cut way down and combined with the next paragraph.

We have cut down this paragraph.

Line 77: Both this paragraph and the one above seem kindof out of place. I understand emphasizing the importance of masking and policies surrounding it, but you spend too much time establishing the importance and not enough time on the next paragraph, tying together Twitter use and masking. It feels disjointed.

We have cut down this paragraph. And added a bit more on social media.

Line 86: The final paragraph should be longer. I like using numbered points, but they should be incorporated into the text better, rather than a separate list. You could even put them as a table in the methods

We have made this paragraph longer.

Line 96: The entire methods section should probably be separated into at least 2 subsections for clarity. I.e. Tweet Sampling, Social network analysis. It’s overall a little wordy for a methods paragraph.

We have split this section into two subsections.

Line 106: “In the network circles” probably needs hyphens for clarity

We have now added a hyphen.

Line 116: Results 3.1: I don’t think you need the first section at all, the top 10 hashtags. It seems pointless, especially explaining the subtle cultural contexts for some of the tags, like beer. I would drop this section entirely.

We respect the authors view, but feel this table highlights the international nature of the discussion.

Line 143: This section needs some major work. First off, structure is spelled wrong in the title. The first sentence is not needed. The main problem is the text is just talking about the figure, as opposed to the figure supporting the text. It also needs way more analysis of the groups in the figure. The entire paragraph is really generic and feels pointless without more analysis. The actual analysis is cool with a cool figure, get into it!

This has been explained in more detail.

Line 151: There is a decided lack of statistical methods for all of these figures. You didn’t define “bridges” in the context of followers. Also, how are you measuring influence? Just because you have followers doesn’t mean you actually influence them. Maybe measuring retweets would be better, and maybe you did do that, but it’s not clear if you did. You also didn’t define betweenness centrality. What is that?

We have added more to this sentence.

Line 159: This paragraph feels very unscientific and pointless. Too many “suggests”, “may” and “might”. It just doesn’t feel informative about, well, anything, it’s just a list of twitter followers.

We have edited the language in this section

Line 169: Once again, not much here to help the reader interpret the data.

We have edited this section

Line 179: “revolved” not “resolved”

This has been done.

Line 181: This last sentence is just unsupported by any data presented. I get that it’s providing some clarity, but its author interpretation, so it should be in the discussion.

This has been moved to the discussion,

Line 187: How were these ranks determined? “The content…appeared to be similar to that of the results of the analysis overall”, How so? What criteria are you using to determine this?

This has been ranked by betweenness centrality.

Line 206: Lose the first sentence, and you don’t need to introduce each topic being discussed, i.e. “in response to the first research question”. The discussion should go back over the main questions and hypotheses from the intro, then go back through the results to show how they addressed those questions, and provide some interpretation.

We have edited this.

Line 236: This paragraph is incomplete and has multiple grammatical and spelling errors. Also, your last reference needs to be updated in the bibliography, it was published August 18.

We have proofread and updated the citation for this.

Round 2

Reviewer 1 Report

A social network analysis of tweets related to masks and COVID-19

Readers of high-impact journals such as IJERPH hope to find scientific contributions (theoretical and / or methodological) to the field of public health. In the case of empirical investigations, they hope that the data collection and analysis methods will have a solid and rigorous scientific basis. Readers also hope that the researchers have contrasted the results.

Unsubstantiated assertions are contrary to the scientific method:

"Social media is a useful platform for raising awareness of various issues, and Twitter is a useful platform for listening to public views and opinions on a range of topics in real-time."

This assertion may be true, but they must cite at least two prestigious papers that have demonstrated it.

"... a week of data in a period when this topic was highly present in social media."

"... USA as one of the countries with more controversy around mask wearing."

In the case of not having statistics or data in the abundant related literature, you can cite some news in prestigious press such as the New York Times or the Washington Post.

Automatic content analysis on social media is tricky, especially on Twitter. Have you verified the reliability of NodeXL? How accurate are the classifications obtained? How do you come to the conclusion: "The first two co-words appear to be encouraging the use of face-masks"? Those word pairs can also be used by pandemic deniers and those who refuse to wear a mask, and you base the main conclusion of your study on those co-occurrences.

In the discussion, it is necessary to compare your results with others obtained in the countless researches on masks and COVID-19 (Google Scholar).

In short, what accuracy or percentage of hits do the results have?    

Author Response

We have made a number of further edits to the manuscript. 

Reviewer 3 Report

The little things like spelling and grammar have been sortof fixed (still see some problems) but the overall issues from all three reviewers have not been addressed.

The figures and tables still have little to no explanation.

The NodeXL references are all from the primary author, and to claim "The software NodeXL has been used in several other published studies and it is a reliable and recognised [sic] tool to analyse [sic] twitter data." is not the most accurate statement. It's not a very common tool.

Statistics have not been addressed, nor a lot of data in general. Why not present the raw data, or at least the betweenness centralities? We're having to just take your word for essentially everything, especially figure 1.

Most of the addressed changes are inadequate.

The overall concept is not bad, honestly, but the paper's approach is not scientific enough. Why not use Twitter's API for more data and analyze with R? You may have used that, but you didn't say. I think one of the biggest limitations is your time limit. The potential data set is massive, with lots of cool conclusions you could tease out, so to limit yourself to such small numbers feels like unreached potential. We need more. I suggest the authors look at this paper: https://www.ncbi.nlm.nih.gov/pmc/articles/PMC7487966/

Author Response

(The authors gave the same response as above.)
